# Serum Adipocyte Fatty-Acid Binding Protein as an Independent Marker of Peripheral Artery Disease in Patients with Type-2 Diabetes Mellitus

**DOI:** 10.3390/ijerph19159459

**Published:** 2022-08-02

**Authors:** Bang-Gee Hsu, Chin-Yee Mah, Du-An Wu, Ming-Chun Chen

**Affiliations:** 1Division of Nephrology, Hualien Tzu Chi Hospital, Buddhist Tzu Chi Medical Foundation, Hualien 97004, Taiwan; gee.lily@msa.hinet.net; 2School of Medicine, Tzu Chi University, Hualien 97004, Taiwan; mcyalyssa@gmail.com (C.-Y.M.); despdu@yahoo.com.tw (D.-A.W.); 3Division of Metabolism and Endocrinology, Hualien Tzu Chi Hospital, Buddhist Tzu Chi Medical Foundation, Hualien 97004, Taiwan; 4Department of Pediatrics, Hualien Tzu Chi Hospital, Buddhist Tzu Chi Medical Foundation, Hualien 97004, Taiwan

**Keywords:** ankle–brachial index, peripheral arterial disease, adipocyte fatty-acid binding protein, diabetes mellitus

## Abstract

The adipocyte fatty-acid binding protein (A-FABP) is predominantly expressed in macrophages and adipocytes and is an essential mediator of inflammation and atherosclerosis pathogenesis. Atherosclerosis is an aggravating factor for peripheral arterial disease (PAD). Our study intended to study the association between PAD and serum A-FABP levels in type-2 diabetes mellitus (T2DM) patients. One hundred and twenty T2DM subjects were enrolled in the study. Fasting blood samples were collected to determine biochemical data and A-FABP levels. By the automatic oscillometric method, the ankle–brachial index (ABI) was measured. Low ABI was defined as any value < 0.9. Twenty participants with T2DM (16.7%) were included in the low ABI group. Low ABI T2DM participants had an increased mean body mass index, body fat mass, systolic blood pressure, C-reactive protein, urine albumin–creatinine ratio, and A-FABP levels compared to those in the normal ABI group. After variables significantly associated with PAD were adjusted by multivariate logistic regression analyses, circulating A-FABP levels (odds ratio [OR]: 1.138; 95 percent confidence interval [CI]: 1.023–1.266; *p* = 0.017) were identified as the independent marker of PAD. In conclusion, fasting serum A-FABP value has positive association with PAD in T2DM patients.

## 1. Introduction

Type-2 diabetes mellitus (T2DM) is associated with various complications. Cardiovascular disease (CVD) causes more than 60% of the life-years lost from DM and is the main cause of disability and mortality among populations with T2DM [1,2]. In DM patients, the most common cardiovascular manifestations include peripheral arterial disease (PAD), coronary heart disease, and heart failure [3]. PAD, a vascular disease caused by systemic atherosclerosis, can induce peripheral ischemia and has a strong association with the increased prevalence of cardiovascular events [4,5]. Ankle–brachial index (ABI) is not only a valuable tool in diagnosing PAD, but is also a surrogate marker of cardiovascular events in the general and diseased populations [6]. DM, an important aggravating factor for PAD, also increases the restenosis risk and can diminish the effectiveness of percutaneous endovascular revascularization in PAD patients [7]. Diabetic patients have a 1.5- to 4.0-fold higher risk of PAD occurrence than individuals without DM [8,9]. PAD in DM usually affects younger patients and is commonly associated with rapid progression, multilevel artery stenosis and occlusions, especially below the knee, as well as serious comorbidities and mortality, compared to that in the general population [10]. A longitudinal cohort study reported by Leibson et al. demonstrated that DM subjects with PAD have a higher mortality rate than subjects with either DM or PAD [11]. A recent review study also stated the prominent association between frailty syndrome in DM patients and PAD, especially in the geriatric populations or subjects after amputation [12].

The adipocyte fatty-acid binding protein (A-FABP, also called FABP4), a 15-kDa member of the intracellular fatty-acid binding protein mainly expressed in adipose cells or macrophages, is associated with arterial stiffness, dysmetabolic syndrome, and CVD [13,14]. In animal models, the proximal aorta atherosclerotic lesion size was ameliorated in A-FABP/apolipoprotein-E double-knockout mice compared to that in apolipoprotein-E-deficient controls [15,16]. Moreover, A-FABP deficiency limited to macrophages markedly reduced the atherosclerotic lesion size in vivo compared to total A-FABP deficiency [16]. In line with these animal study results, a ten-year prospective study revealed that circulating A-FABP concentration was the main pathophysiological mediator of atherosclerosis and had a positive association with the long-term outcome in coronary artery disease (CAD) populations [17].

The PAD prevalence of DM patients varies from 20% to 30% depending on the research population [18]. Cross-sectional studies have revealed a correlation between A-FABP expression and T2DM. The A-FABP value is positively associated with microalbuminuria even in T2DM populations with preserved renal function [19]. Our previous publications also demonstrated that the A-FABP level had a positive association with arterial stiffness in patients who had undergone renal transplant and were on hemodialysis [20,21]. However, publications on the A-FABP level and PAD among populations with DM are lacking. Therefore, we evaluated various predictors of PAD development and examined their correlation with the circulating A-FABP level in participants with T2DM. We also examined whether elevated serum A-FABP concentration was independently associated with PAD using the ABI.

## 2. Materials and Methods

### 2.1. Study Population

From November 2014 to March 2015, 140 patients with T2DM were enrolled at a medical center in Hualien, Taiwan. Twenty participants were excluded due to the use of protease-activated receptor-1 antagonists (*n* = 5), or warfarin (*n* = 3), or acute infection (*n* = 2) during blood sampling, and refusal to offer informed consent (*n* = 10). Finally, 120 patients with T2DM were invited to this study. The Research Ethics Committee of Hualien Tzu Chi Hospital approved this study (IRB103-136-B). Every participant provided written informed consent before entering this study. In the morning, after the participants had sat for ten minutes, the blood pressure (BP) was determined by standard mercury sphygmomanometers. Systolic BP (SBP) and diastolic BP (DBP) were obtained every 5 min, 3 times, and averaged for further analyses. Hypertension was defined as BP values ≥ 140/90 mmHg, or using any antihypertensive medication over the past two weeks.

### 2.2. Anthropometric and Body Composition Analysis

The height and body weight of each patient were recorded to the 0.50 cm and 0.50 kg, respectively. We calculated the body mass index (BMI) as a ratio of weight and height squared (kg/m^2^). Body fat mass was measured by bioimpedance analysis (Biodynamic-450; Biodynamics Corporation, Seattle, USA) in accordance with the standard tetrapolar whole-body (hand-foot) technique [13,22].

### 2.3. Biochemical Analyses

An overnight-fasting 10 mL blood sample from each participant was collected and immediately centrifuged (3000× *g*, 10 min). Serum fasting glucose, glycated hemoglobin (HbA1c), total cholesterol, triglycerides (TG), high-density lipoprotein cholesterol (HDL-C), low-density lipoprotein cholesterol (LDL-C), blood urea nitrogen (BUN), creatinine, and C-reactive protein (CRP) levels were measured using a Siemens Advia 1800 autoanalyzer (Siemens Healthcare GmbH, Henkestr, Germany) [13,22,23]. In a random spot urine sample, we tested urine albumin–creatinine ratio (UACR). We measured serum A-FABP concentrations via commercially available enzyme immunoassay kits (SPI-BIO, Montigny-le-Bretonneux, France) [13,22]. The intra- and inter-assay coefficients of variation were 6.6% and 5.1%, respectively. Using the Chronic Kidney Disease Epidemiological Collaboration (CKD-EPI) equation, the estimated glomerular filtration rate was calculated [24].

### 2.4. Ankle–Brachial Index Measurements

We measured ABI values with an automatic machine (VaSera VS-1000, Fukuda Denshi Co., Ltd., Tokyo, Japan) which recorded both the arm and ankle BP simultaneously with the oscillometric method [23]. The occlusive monitoring cuffs were tightly placed on four limbs, and heart sound and electrocardiographic data were recorded in the rest supine position for 10 min during the ABI measurement. The ABI value was determined as the SBP ratio of ankle to brachial, while the lower ankle SBP value was used for data analysis. We recorded the parameters for both lower extremities of each patient repeatedly and calculated the mean value. An ABI threshold of 0.9 has been used for PAD diagnosis in a previous study [25]. In this study, the low ABI group consisted of patients with a right or left ABI value < 0.9, and those whose values ≥ 0.9 were defined as the control group, as in previous work [23].

### 2.5. Statistical Analysis

The Kolmogorov–Smirnov test was used to determine whether the data are normally distributed. Continuous variables with normal distribution were shown as the mean ± standard deviation and were compared by using Student’s *t*-test. Non-normally distributed variables were reported by the median (interquartile range) and examined via the Mann–Whitney U test (BUN, creatinine, fasting glucose, HbA1c, TG, and UACR). We used the χ^2^ test for categorical variables analysis. Variables significantly associated with PAD among T2DM populations were tested for multivariate logistical regression analysis (adopted factors: BMI, body fat mass, SBP, CRP, UACR, and A-FABP). The association between A-FABP concentrations and clinical variables was evaluated using the nonparametric Spearman’s rank correlation coefficient. The area under the receiver operating characteristic (ROC) curve was analyzed to identify the A-FABP level that predicts PAD in T2DM patients. We used SPSS Statistics 19.0 software for data analyses. A *p*-value of 0.05 or lower was considered significant. 

## 3. Results

The demographic, clinical, and biochemical information of the 120 enrolled patients with T2DM is shown in Table 1. In total, 75 males and 45 female participated in the study, and 58 patients had hypertension (48.3%). Typical risk factors for CVD and atherosclerosis were highly prevalent in our study population. The low ABI group comprised 20 patients with T2DM (16.7%). DM subjects within the low ABI group had higher BMI (*p* = 0.037), body fat mass (*p* = 0.019), SBP (*p* = 0.002), CRP (*p* < 0.001), UACR (*p* = 0.001), and A-FABP (*p* < 0.001) values than those within the normal ABI group. Medications used by the study participants included statins (*n* = 56; 46.7%), fibrates (*n* = 4; 3.3%), metformin (*n* = 66; 55.0%), sulfonylurea (*n* = 68; 56.7%), dipeptidyl peptidase-4 inhibitor (*n* = 71; 59.2%), and insulin (*n* = 31; 25.8%). No statistically significant differences existed between the patients with low and normal ABI values regarding sex, comorbid conditions with hypertension, the use of anti-hyperlipidemic medications, or the anti-diabetic medications usage, such as statins, fibrates, metformin, sulfonylureas, dipeptidyl peptidase-4 inhibitors, and insulin.

Adjustment was made for factors significantly associated with PAD (BMI, body fat mass, SBP, CRP, UACR, and A-FABP) in the multivariate logistic regression analysis. The analysis showed that elevated circulating A-FABP level (odds ratio [OR]: 1.138; 95% confidence interval [CI]: 1.023–1.266; *p* = 0.017), SBP (OR: 1.041; 95% CI: 1.004–1.079; *p* = 0.028), and CRP level (per 0.1 mg/dL increment, OR: 1.275; 95% CI: 1.067–1.523; *p* = 0.008) were risk factors for PAD in T2DM subjects (Table 2). The ROC curve diagram for the PAD prediction represented that the area under the ROC curve for the parameter A-FABP level was 0.823 (95% CI, 0.743–0.887; *p* < 0.001; Figure 1).

Table 3 shows the correlation between serum A-FABP and clinical variables by Spearman’s correlation analysis. First, left, and right ABI were negative correlated with A-FABP (*r* = –0.418 and –0.474, respectively, both *p* < 0.001). Second, serum A-FABP was found to be positively correlated with BMI (*r* = 0.271, *p* = 0.003), body fat mass (*r* = 0.379, *p* < 0.001), SBP (*r* = 0.249, *p* = 0.006), TG (*r* = 0.319, *p* < 0.001), CRP (*r* = 0.382, *p* < 0.001), and UACR (*r* = 0.362, *p* < 0.001) but negatively associated with HDL-C (*r* = −0.186, *p* = 0.006), and eGFR (*r* = −0.442, *p* < 0.001). 

## 4. Discussion

Our cross-sectional study of 120 patients with T2DM showed that serum A-FABP concentrations, together with SBP and CRP, were independent predictors of PAD diagnosed based on ABI values.

Systemic atherosclerosis events are the most important factor determining PAD prevalence and are increasing in number. It is estimated that 200 million patients are affected by PAD around the world [26]. PAD can be detected by various methods; however, ABI measurement is the noninvasive, standard approach for the diagnosis of PAD [27]. In the general population, ABI values less than 0.9 has been regarded as a predictor to diagnose PAD with 95% sensitivity [28]. PAD has high prevalence and morbidity rates and is characterized by high risk of CVD, amputation, and even death [5].

Studies have identified many established risk factors for PAD development, including smoking, DM, hypertension, obesity, dyslipidemia, aging, and chronic kidney disease [8,29,30], all of which induce a chronic inflammatory response, thereby aggravating the progression of atherosclerosis. DM is the main risk factor for PAD, and according to the Framingham Heart Study, twenty percent of patients with PAD symptoms are diagnosed with DM [31]. A positive association has been indicated between obesity and PAD in both the general population and hemodialysis patients [32], and it has even been reported that obesity is associated with PAD severity [33]. Previous researchers have also found that SBP value is significantly increased in PAD compared to non-PAD groups [34]. This finding likely stems from changes in the Windkessel effect, which explains arterial blood perfusion toward the periphery. With the progression of atherosclerosis, hardening of the vascular wall leads to elevated SBP and reduces the Windkessel effect [35]. Recently, a meta-analysis reported that even mild-to-moderate chronic kidney disease might increase the risk of PAD. Besides, UACR significantly improves PAD risk discrimination and prediction of amputation in addition to traditional parameters [30]. In line with these publications, our current study also provides clinical evidence that higher BMI, body fat mass, SBP, and UACR values in T2DM patients with low ABI are associated with PAD.

PAD reflects systemic atherosclerosis, a complicated process involving oxidative stress, endothelial dysfunction, vascular smooth muscle cell activation, changes in matrix metabolism, platelet activation, and lipid disturbances [36]. The ABI, a widely used, non-invasive, standard tool for PAD diagnosis, is a reliable and highly reproducable method to detect asymptomatic PAD in the early stage. Low ABI values are associated with many CVD risks in the general population and T2DM subjects [6]. A-FABP is traditionally regarded as a biomarker expressed predominantly in adipose tissue and macrophages. It is derived from monocytes activated by PPAR agonists and oxidized LDL. A-FABP is an important mediator of inflammation and is involved in the pathogenesis of obesity-related insulin resistance, as well as the development of metabolic syndrome and T2DM [37]. Previous reports demonstrated that elevated circulating A-FABP values positively correlated with the low HDL-C level and high BMI, waist circumference, BP, total cholesterol, TG, LDL-C, insulin levels, and homeostasis model assessment-estimated insulin resistance (HOMA-IR) values [38]. Our present study also revealed that A-FABP was positively correlated with BMI, body fat mass, SBP, triglyceride, CRP, and UACR but negatively associated with HDL-C and eGFR. This may explain the effect of A-FABP leading to metabolic deterioration, thereby aggravating atherosclerosis [39]. Results from pharmacological intervention studies in animal models indicate that the serum A-FABP level is a central player in inducing insulin resistance, increasing plasma lipid levels, and promoting the formation of vulnerable atherosclerotic plaques via multiple mechanisms such as autocrine or systemic endocrine pathways in macrophages [40]. A study using different mouse models demonstrated that a specific orally administered A-FABP antagonist was highly effective in the T2DM and atherosclerosis treatment [41]. Recent publications in humans also revealed an association between A-FABP and atherosclerotic diseases and the inflammatory status of the arterial system [42,43]. Administration of an A-FABP antagonist also contributes to the elimination and reduced development of atherosclerotic plaques in vitro and in vivo [41]. A-FABP has been linked to increased atherosclerotic plaque burden in CAD patients and has a positive association with the intima–media thickness of carotid artery in Chinese adults, as assessed by ultrasound [44,45]. Likewise, the study by Bao and his colleagues revealed that circulating A-FABP level is positively correlated with the severity of atherosclerotic disease and a prominent aggravating factor for the CAD occurrence in Chinese women [46]. Inflammation measured by CRP levels as a parameter of PAD occurrence has been reported in the inter-scientific consensus on PAD management [42]. Consistent with prior published data, we found that CRP concentrations have a positive correlation with low ABI values in T2DM patients. Serum CRP represents the primary cause of increased inflammation, which may be a causal predictor of plaque formation and arterial atherosclerotic disease. Our study revealed that circulating A-FABP and CRP levels were independent predictors of PAD in T2DM populations in multivariate logistic regression analyses even after adjusting for variables significantly associated with PAD. Moreover, a significant negative correlation between A-FABP and bilateral ABI values was noted by Spearman’s correlation analysis. These data indicate that A-FABP-related inflammatory responses may aggravate atherosclerotic progression in PAD. Thus, A-FABP levels could be a more specific PAD marker to evaluate ABI in patients with T2DM [47].

Our research has limitations. First, this research was conducted as a single-center, cross-sectional study without the control group, and various lifestyle habits that may affect PAD occurrence, such as smoking and alcohol consumption, were not recorded [48,49]. This lack of information may bias our research, and further longitudinal studies are required to validate our results. Second, the sample of 120 patients with DM in one medical center does not represent of the whole population of Taiwan or all ethnicities. Although a previous study by Ding et al. also investigated the relationship between circulating A-FABP levels and PAD in 488 Chinese T2DM subjects with a larger sample size than our present study, they used ultrasound examination for diagnosing PAD stenosis [47]. Operator-dependent ultrasound data may influence the study result. Our study used ABI, the widely used and valuable diagnostic tool for PAD diagnosis, which is more reproducible and reliable than ultrasound examination and can detect PAD in the early stage. Further studies with larger sample sizes including hospitals across multiple cities or even multiple countries are warranted to establish a more definite conclusion. Third, according to review articles, antilipidemic agents like statins and new antidiabetic drugs such as DPP4i decrease the prevalence of arterial stiffness and cardiovascular risk [50,51]. However, no significant differences between groups with low and normal ABI values were found in our study in relation to the drugs used, though this may be related to the small sample size and poor drug compliance of some patients. Thus, further investigations of the relationship between the abovementioned medications and PAD in patients with T2DM are needed. Fourth, although the ABI plays a vital role in the diagnosis of PAD, ABI may be falsely elevated due to calcification of the artery’s middle layer wall in the DM population. Some studies had recommended using the to–-brachial index (TBI) instead of the ABI for the PAD diagnosis in DM patients [12]. However, the TBI value of PAD diagnosed among DM subjects is still unclear. More research is needed to evaluate the application of TBI in the DM population. 

## 5. Conclusions

Our present study showed, for the first time, that the fasting serum A-FABP level is positively associated with PAD. Moreover, in T2DM populations, fasting serum A-FABP levels are regarded as a consistent and significant predictor associated with PAD determined by ABI.

## Figures and Tables

**Figure 1 ijerph-19-09459-f001:**
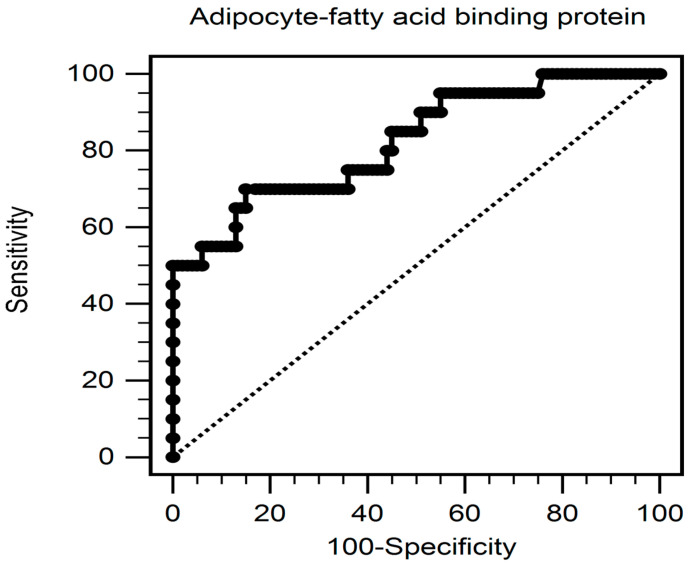
The area under the receiver operating characteristic curve represents the diagnostic power of the adipocyte fatty-acid binding protein value for predicting peripheral artery disease in type-2 diabetes mellitus subjects.

**Table 1 ijerph-19-09459-t001:** Clinical variables at low ABI and control group in 120 type-2 diabetes mellitus patients.

Variables	All Patients (*n* = 120)	Control Group (*n* = 100)	Low ABI Group (*n* = 20)	*p* Value
Age (years)	61.50 ± 12.74	61.32 ± 11.96	62.40 ± 16.38	0.731
Height (cm)	162.97 ± 8.27	163.23 ± 8.40	161.68 ± 7.63	0.444
BW (kg)	71.88 ± 14.35	71.10 ± 13.36	75.75 ± 18.49	0.188
Body mass index (kg/m^2^)	26.94 ± 4.25	26.58 ± 3.90	28.75 ± 5.45	0.037 *
Body fat mass (%)	30.23 ± 7.28	29.54 ± 7.21	33.70 ± 6.75	0.019 *
Left ABI	1.06 (1.00–1.13)	1.09 (1.03–1.14)	0.89 (0.82–0.90)	<0.001 *
Right ABI	1.08 (1.02–1.14)	1.10 (1.06–1.15)	0.92 (0.88–0.96)	<0.001 *
SBP (mmHg)	139.31 ± 18.87	136.92 ± 16.85	151.25 ± 23.89	0.002 *
DBP (mmHg)	81.32 ± 10.25	80.77 ± 9.36	84.05 ± 13.86	0.193
Total cholesterol (mg/dL)	160.54 ± 30.39	159.29 ± 29.51	166.80 ± 34.58	0.315
Triglyceride (mg/dL)	109.00 (78.50–182.00)	110.50 (75.50–187.25)	103.00 (86.75–172.50)	0.863
HDL-C (mg/dL)	46.30 ± 12.36	46.43 ± 12.37	45.65 ± 12.63	0.798
LDL-C (mg/dL)	99.17 ± 26.99	98.66 ± 25.67	101.70 ± 33.50	0.648
Fasting glucose (mg/dL)	138.00 (119.50–171.75)	138.00 (121.00–175.00)	136.50 (102.00–169.00)	0.481
Glycated hemoglobin (%)	7.20 (6.50–9.00)	7.20 (6.50–8.88)	7.25 (6.15–9.85)	0.972
BUN (mg/dL)	15.00 (12.00–18.00)	15.00 (12.00–18.00)	18.00 (12.25–19.00)	0.282
Creatinine (mg/dL)	0.8 (0.70–1.00)	0.80 (0.70–1.00)	0.80 (0.70–0.90)	0.718
eGFR (mL/min)	91.12 ± 26.39	91.34 ± 26.00	90.00 ± 28.93	0.837
C-reactive protein (mg/dL)	0.08 (0.05–0.23)	0.06 (0.05–0.15)	0.51 (0.11–1.02)	<0.001 *
UACR (mg/g)	14.00 (7.13–54.27)	12.19 (6.06–33.55)	67.68 (22.99–182.07)	0.001 *
A-FABP (ng/mL)	21.80 ± 9.02	19.88 ± 6.70	31.44 ± 12.61	<0.001 *
Male, *n* (%)	75 (62.5)	65 (65.0)	10 (50.0)	0.206
Hypertension, *n* (%)	58 (48.3)	46 (46.0)	12 (60.0)	0.253
Statin usage, *n* (%)	56 (46.7)	45 (45.0)	11 (55.0)	0.413
Fibrate usage, *n* (%)	4 (3.3)	4 (4.0)	0 (0)	0.363
Metformin usage, *n* (%)	66 (55.0)	56 (56.0)	10 (50.0)	0.622
Sulfonylurea usage, *n* (%)	68 (56.7)	54 (54.0)	14 (70.0)	0.187
DDP-4 inhibitor usage, *n* (%)	71 (59.2)	60 (60.0)	11 (55.0)	0.678
Insulin usage, *n* (%)	31 (25.8)	26 (26.0)	5 (25.0)	0.926

Continuous variables are shown as the mean ± standard deviation and tested by Student’s independent *t*-test. Non-normally distributed data were reported by the median (interquartile range); they were tested by the Mann–Whitney U test. Categorical variables were examed by the χ^2^ test and are reported as numbers (%). ABI, ankle–brachial index; BW, body weight; SBP, systolic blood pressure; DBP, diastolic blood pressure; HDL-C, high-density lipoprotein cholesterol; LDL-C, low-density lipoprotein cholesterol; BUN, blood urea nitrogen; eGFR, estimated glomerulus filtration rate; UACR, urine albumin–creatinine ratio; A-FABP, adipocyte fatty-acid binding protein; DDP-4, dipeptidyl peptidase-4. * A *p*-value of 0.05 or lower was considered significant.

**Table 2 ijerph-19-09459-t002:** Multivariate logistic regression analysis of the variables associated with peripheral arterial disease in 120 patients with diabetes mellitus.

Variables	Odds Ratio	95% Confidence Interval	*p* Value
Adipocyte fatty-acid binding protein, 1 ng/mL	1.138	1.023–1.266	0.017 *
Systolic blood pressure, 1 mmHg	1.041	1.004–1.079	0.028 *
C-reactive protein, 0.1 mg/dL	1.275	1.067–1.523	0.008 *
Body mass index, 1 kg/m^2^	0.974	0.808–1.174	0.780
Body fat mass, 1 %	1.012	0.912–1.123	0.823
Urine albumin-to-creatinine ratio, 1 mg/g	0.999	0.998–1.001	0.287

Data were analyzed using multivariate logistic regression (adopted factors: body fat mass, body mass index, C-reactive protein, systolic blood pressure, urine albumin–creatinine ratio, and adipocyte fatty-acid binding protein). * *p* < 0.05 was statistically significant.

**Table 3 ijerph-19-09459-t003:** Spearman correlation coefficients between adipocyte fatty-acid binding protein and clinical variables in 120 type-2 diabetes mellitus patients.

Variables	Spearman Coefficient of Correlation	*p* Value
Age (years)	0.088	0.340
Body mass index (kg/m^2^)	0.271	0.003 *
Body fat mass (%)	0.379	<0.001 *
Left ankle–brachial index	–0.418	<0.001 *
Right ankle–brachial index	−0.474	<0.001 *
Systolic blood pressure (mmHg)	0.249	0.006 *
Diastolic blood pressure (mmHg)	0.116	0.208
Total cholesterol (mg/dL)	0.081	0.378
Triglyceride (mg/dL)	0.319	<0.001 *
HDL-C (mg/dL)	–0.186	0.042 *
LDL-C (mg/dL)	0.048	0.606
Fasting glucose (mg/dL)	0.165	0.072
eGFR (mL/min)	–0.249	0.006 *
C-reactive protein (mg/dL)	0.382	<0.001 *
UACR (mg/g)	0.362	<0.001 *

Analysis of data was performed using the Spearman correlation analysis. HDL-C, high-density lipoprotein cholesterol; LDL-C, low-density lipoprotein cholesterol; eGFR, estimated glomerulus filtration rate; UACR, urine albumin–creatinine ratio. * *p* < 0.05 was considered statistically significant.

## Data Availability

All data which support the findings of our study are included in this article.

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
