# Peer review of "Serum Adipocyte Fatty-Acid Binding Protein as an Independent Marker of Peripheral Artery Disease in Patients with Type-2 Diabetes Mellitus"

_ijerph, 2022, doi:10.3390/ijerph19159459_

Round 1

Reviewer 1 Report

Diabetes mellitus type 2 is a world health challenge. The number of patients living with diabetes has tripled since 2000, and preventive, diagnostic, and therapeutic interventions are urgent. In the present manuscript, Hsu et al. measured the ankle-brachial index of diabetic patients and grouped them into low and high ABI. Many biochemical parameters were analyzed. Patients with low ABI had increased serum A-FABP levels, and it was proposed as marker of PAD.

Although the work is of interest, and new diagnosis methods are necessary, the work presents technical limitations. The sample size is small, and the work is not original. Similar results were introduced in the manuscript “Serum adipocyte fatty acid-binding protein levels are associated with peripheral arterial disease in women, but not men, with type 2 diabetes mellitus” by Ding M and colleagues in 2018 in a study with 488 patients. Additionally, the work is biased once the group does not reflect the entire population. The findings presented have potential significance, but the group needs to be increased and reflect the whole population.

Author Response

Response to Reviewer 1: 

We thank the reviewers for their helpful comments. We have revised the manuscript according to the reviewers' comments/suggestions, and have provided an itemized response in the following sections.

Diabetes mellitus type 2 is a world health challenge. The number of patients living with diabetes has tripled since 2000, and preventive, diagnostic, and therapeutic interventions are urgent. In the present manuscript, Hsu et al. measured the ankle-brachial index of diabetic patients and grouped them into low and high ABI. Many biochemical parameters were analyzed. Patients with low ABI had increased serum A-FABP levels, and it was proposed as marker of PAD.

Although the work is of interest, and new diagnosis methods are necessary, the work presents technical limitations. The sample size is small, and the work is not original. Similar results were introduced in the manuscript “Serum adipocyte fatty acid-binding protein levels are associated with peripheral arterial disease in women, but not men, with type 2 diabetes mellitus” by Ding M and colleagues in 2018 in a study with 488 patients. Additionally, the work is biased once the group does not reflect the entire population. The findings presented have potential significance, but the group needs to be increased and reflect the whole population.

Response: Thanks for the comments from the Reviewer. Our study was a single-center, cross-sectional study with the limited sample size. We had stated this in the 5th paragraph of our Discussion part as our study limitation as " First, this research was conducted as a single-center, cross-sectional study without the control group, and various lifestyle habits that may affect PAD occurrence, such as smoking and alcohol consumption, were not recorded [43, 44]. This lack of information may bias our research, and further longitudinal studies are required to validate our results. Second, the sample of 120 patients with DM in one medical center does not represent of the whole population of Taiwan or all ethnicities.” We agree the Reviewers’ comment and had stated the direction of our future study as " further studies with larger sample sizes including hospitals across multiple cities or even multiple countries are warranted to establish a more definite conclusion." in our limitation. In addition, the study “Serum adipocyte fatty acid-binding protein levels are associated with peripheral arterial disease in women, but not men, with type 2 diabetes mellitus” by Ding M and colleagues in 2018 investigated the relationship between serum A-FABP concentrations and PAD in 488 Chinese patients with T2DM. This study used ultrasound examination for peripheral arterial stenosis evaluation by a single trained ultrasound practitioner. Although the sample size was larger than our present study, operator dependent exam data may have influence on the study result. Our study used ABI, the widely used valuable diagnostic tool for PAD diagnosis, is more reliable and high reproducibility than ultrasound examination alone and can detect PAD in early stage. We had also added above information to our limitation at 5th paragraph of our Discussion part.

Reviewer 2 Report

Hsu et al. prepared an interesting research article in which they proposed serum adipocyte fatty acid-binding protein as a potential new biomarker of peripheral arterial disease (PAD) in patients with type 2 diabetes mellitus (T2DM). The subject plays a crucial role in contemporary medicine because T2DM is called the epidemic of the 21st century; on the other hand, cardiovascular diseases (CVDs) remain the leading cause of morbidity and mortality in patients with T2DM. Improving the knowledge about pathogenesis and new diagnostic tools for CVDs in patients with T2DM is therefore very important. The paper is generally well prepared, but I would like to give some recommendations that, in my opinion, may further improve the quality of the manuscript.

The “introduction” should be improved, and some accessory aspects should be mentioned. Although the information on the adipocyte fatty acid-binding protein and its presumed association with PAD in patients in T2DM is well described, basic details on CVD in the course of T2DM should be extended. Endothelial dysfunction is one of the most critical mechanisms in the early phase of atherosclerosis development. It would be worth mentioning that in patients with PAD in the course of DM, arteries below the knee are more often affected, and stenoses and occlusions are more likely to be multilevel compared to the general population. DM is a risk factor for restenosis, which can diminish the effectiveness of endovascular revascularization. The relationship between PAD and Frailty syndrome in patients with diabetes is worth mentioning. It should be noted that ABI is not only a valuable diagnostic tool in diagnosing PAD but also a marker of cardiovascular events risk. In patients with DM, ABI may be falsely elevated due to calcification of the middle layer of the artery’s wall. You can use, for example, the following high-quality and up-to-date positions of scientific literature: doi.org/10.3390/ijerph17249339; doi.org/10.3390/ijerph182211970; doi.org/10.4239/wjd.v6.i7.961; doi.org/10.1155/2018/1098039; doi.org/10.1155/2020/6471098; doi.org/10.3390/jcm8060870.

Material and methods are well described. I have no additional suggestions for this part of the manuscript.

Below table 1, there is a list of the abbreviations explained. Please note that in some cases, the abbreviation precedes the description, and in others, it is the description followed by the abbreviation. This should be harmonized.

Discussion is generally well structured. In my opinion, in limitations of the study should be paid attention to possible falsely elevated ABI value in the population with diabetes and lack of TBI measurement.

Author Response

Response to Reviewer 2: 

We thank the reviewers for their helpful comments. We have revised the manuscript according to the reviewers' comments/suggestions and provided an itemized response in the following sections.

The “introduction” should be improved, and some accessory aspects should be mentioned. Although the information on the adipocyte fatty acid-binding protein and its presumed association with PAD in patients in T2DM is well described, basic details on CVD in the course of T2DM should be extended. Endothelial dysfunction is one of the most critical mechanisms in the early phase of atherosclerosis development. It would be worth mentioning that in patients with PAD in the course of DM, arteries below the knee are more often affected, and stenoses and occlusions are more likely to be multilevel compared to the general population. DM is a risk factor for restenosis, which can diminish the effectiveness of endovascular revascularization. The relationship between PAD and Frailty syndrome in patients with diabetes is worth mentioning. It should be noted that ABI is not only a valuable diagnostic tool in diagnosing PAD but also a marker of cardiovascular events risk. In patients with DM, ABI may be falsely elevated due to calcification of the middle layer of the artery’s wall. You can use, for example, the following high-quality and up-to-date positions of scientific literature: doi.org/10.3390/ijerph17249339;doi.org/10.3390/ijerph182211970; doi.org/10.4239/wjd.v6.i7.961; doi.org/10.1155/2018/1098039; doi.org/10.1155/2020/6471098; doi.org/10.3390/jcm8060870.

Response: Thanks for the Reviewer’s suggestion. We had added this information to our first paragraph of our Introduction part and cited these four excellent studies to our references (references 6, 7, 10 and 12) as your suggestion.

Material and methods are well described. I have no additional suggestions for this part of the manuscript.

Response: Thanks for the comment from the Reviewer.

Below table 1, there is a list of the abbreviations explained. Please note that in some cases, the abbreviation precedes the description, and in others, it is the description followed by the abbreviation. This should be harmonized.

Response: Thanks for the reminder. We have adjusted the words according to the Reviewer’s suggestion.

Discussion is generally well structured. In my opinion, in limitations of the study should be paid attention to possible falsely elevated ABI value in the population with diabetes and lack of TBI measurement.

Response: Thanks for pointing this out. We had added above limitation at 5th paragraph of our Discussion part as " Fourth, although the ABI plays a vital role in the diagnosis of PAD, ABI may be falsely elevated due to calcification of the artery’s middle layer wall in the DM population. Some studies had recommended using the toe-brachial index (TBI) instead of the ABI for the PAD diagnosis in DM patients [12]. However, the TBI value of PAD diagnosed among DM subjects is still unclear. More research is needed to evaluate the application of TBI in the DM population."

Round 2

Reviewer 1 Report

Based on the previous comments, the authors performed improvements in the new version. However, they were not sufficient for publication.

Author Response

Response: Thanks for the comments from the Reviewer.

    Previous reports have revealed a correlation between A-FABP expression and T2DM. The A-FABP level had a positive association with arterial stiffness in patients who had undergone the renal transplant and were on hemodialysis in our prior studies. However, publications on the A-FABP level and PAD among populations with DM are lacking. Therefore, our present study examined whether elevated serum A-FABP concentration was independently associated with PAD using the ABI value.
   Serum A-FABP was quantified using a commercially available ELSA (SPI-BIO, Montigny-le-Bretonneux, France) according to the manufacturer’ s instructions. The intra- and inter-assay coefficients of variation were 6.6% and 5.1%, respectively. This method has high consistency for serum A-FABP evaluation. We had added this information to Materials and Methods 2.3. Biochemical Analyses part.

    The presence of low ABI, defined as <0.9, has been proposed as a modifier of total cardiovascular risk. This association has been assessed in the general population; in persons with diabetes; and in persons with the previous history of coronary artery disease, in addition to diabetes mellitus and traditional risk factors. Although our study was a single-center, cross-sectional study with a limited sample size, we used the ABI, the widely used noninvasive, standard diagnostic tool for PAD diagnosis. In addition, the ABI value is more reliable and has high reproducibility than ultrasound examination alone, and can detect asymptomatic PAD in the early stage. We had stated this in the 4th and 5th paragraph of our Discussion part and had stated the direction of our future study as " further studies with larger sample sizes including hospitals across multiple cities or even multiple countries are warranted to establish a more definite conclusion." in our limitation as Reviewer’s comment.

Moreover, we had added the table 3 about the Spearman correlation coefficients between A-FABP and clinical variables in our 120 T2DM patients. Our present study revealed that A-FABP was positively correlated with BMI, body fat mass, SBP, triglyceride, CRP, and UACR but negatively associated with HDL-C and eGFR. In addition, a significant negative correlation between A-FABP and bilateral ABI values was noted. These data indicate once again that A-FABP-related inflammatory responses may aggravate atherosclerotic progression in PAD. Thus, A-FABP levels could be a more specific PAD marker evaluated with ABI in patients with T2DM.

Although the sample size of our study is not large, we had used the suitable study method to evaluate the association between serum A-FABP concentration in T2DM patients with PAD using the ABI value. We believe our study has its clinical implications and is worth reporting after the above adjustment.

Reviewer 2 Report

The paper has been improved. I recommend it for publication in its present form. 

Author Response

Response: Thanks for the comment from the Reviewer. 
